# *Vibrio alginolyticus* growth kinetics and the metabolic effects of iron

William A. Norfolk,[1] Charlyn Shue,[1] W. Matthew Henderson,[2] Donna A. Glinski,[2] Erin K. Lipp[1]

**ABSTRACT** *Vibrio alginolyticus* is a naturally occurring marine bacterium, recognized as an emerging pathogen in humans and animals and the second most common cause of vibriosis in the U.S. However, information regarding the physiology and growth of this species in the environment is limited. Here we evaluated the effects of temperature, salinity, and iron condition on the growth response across unique *V. alginolyticus* strains. A combination of growth kinetics and gas chromatography-mass spectrometry-based metabolomics studies was used to evaluate the optimal and tolerable ranges of growth and to characterize the metabolic effects of iron supplementation. All *V. alginolyticus* strains tested demonstrated broad temperature and salinity tolerance, resulting in growth at all measured temperatures (24°C–40°C) and salinities between 1% and 6% (wt/vol) NaCl with optimal growth between 30°C–36°C and 2%–4% NaCl. Environmental strains showed no growth limitation at iron concentrations ranging from 0.5- to 20.0-µM ferric Fe but demonstrated reduced growth at 0.2 µM. Likewise, the number of significantly upregulated metabolites in *V. alginolyticus* cultures grown in iron-replete (4-µM) media was greater than that in iron-deficient (~0 µM) media but varied with prior growth conditions. Detected compounds were associated with key metabolic pathways, namely, amino acid, carbohydrate, lipid, and nucleotide metabolism, suggesting that introduced iron facilitated broad activation of *V. alginolyticus* metabolism and helped to promote growth responses. Combined, these results demonstrate that *V. alginolyticus* strains are capable of rapid growth under a broad range of favorable temperature and salinity levels, which can be affected by the presence of iron.

**IMPORTANCE** Transmission of *V. alginolyticus* occurs opportunistically through direct seawater exposure and is a function of its abundance in the environment. Like other *Vibrio* spp., *V. alginolyticus* are considered conditionally rare taxa in marine waters, with populations capable of forming large, short-lived blooms under specific environmental conditions, which remain poorly defined. Prior research has established the importance of temperature and salinity as the major determinants of *Vibrio* geographical and temporal range. However, bloom formation can be strongly influenced by other factors that may be more episodic and localized, such as changes in iron availability. Here we confirm the broad temperature and salinity tolerance of *V. alginolyticus* and demonstrate the importance of iron supplementation as a key factor for growth in the absence of thermal or osmotic stress. The results of this research highlight the importance of episodic iron input as a crucial metric to consider for the assessment of *V. alginolyticus* risk.

**KEYWORDS** *Vibrio alginolyticus*, iron, physiology, metabolomics, growth kinetics, tolerance

*V*ibrio alginolyticus is a ubiquitous marine bacterium native to coastal and estuarine waters worldwide. As an opportunistic pathogen, this species is an important agent

Address correspondence to Erin K. Lipp, elipp@uga.edu.

The authors declare no conflict of interest.

of both human and animal disease affecting a broad range of host species including marine fishes (1, 2), crustaceans (3, 4), mollusks (5), echinoderms (6), corals (7, 8), marine mammals (9), sea turtles (10), and humans (11, 12). Animal infections have been widely described in association with the aquaculture industry and range in severity with disease signs manifesting as mild epidermal lesions (9) to systemic organ dysfunction and hemorrhage often leading to mass mortality (1, 4). Reported human infections, which have doubled in the U.S. from 2009 to 2019 (13), are strongly associated with recreational and/or occupational exposure to seawater and manifest primarily as opportunistic infections of the ears and pre-existing or sustained wounds (11, 12). While often severe in aquaculture settings [i.e., causing a higher percentage of mortality in affected fish and crustaceans (1, 4)], human infections are typically non-life threatening, presenting as self-limiting or readily treatable through the administration of antibiotics such as ciprofloxacin and tetracycline (14, 15). However, infections in immunocompromised patients have been shown to progress to invasive conditions such as bacteremia and sepsis, greatly increasing the chance of mortality (11, 16). Collectively, the burden of *V. alginolyticus* infections imposes a substantial economic and regulatory encumbrance to aquaculture and public health, with annual cost estimates in excess of one million dollars (USD) for the treatment of human infections in the United States (17, 18) and a global estimated cost of three billion dollars (USD) for the treatment or culling of *Vibrio* aquaculture outbreaks (of which *V. alginolyticus* is a major contributor) (19).

Environmental factors that enhance or inhibit the growth of *V. alginolyticus* populations in the environment are critical to the estimation of exposure risk for this species. Prior studies have shown that temperature and salinity are the two leading environmental determinants of growth for most *Vibrio* species (20, 21) and that increased temperature positively correlates with increased *Vibrio* abundance (20, 22–25). This correlation has been corroborated for *V. alginolyticus* specifically (26, 27) and provides a mechanism for the strong seasonality of infections associated with warmer months (12, 28). *V. alginolyticus* can tolerate temperatures ranging from 5°C to 42°C with faster growth typically occurring between 22°C and 37°C and optimal growth (fastest growth rate) at 35°C ± 2°C (29–33). Second to temperature, salinity is a critical factor for the establishment of *Vibrio* range with species-specific optimal growth occurring from 0 to 35. *V. alginolyticus* has been shown to be tolerable of a wide salinity range from 0.5 to 60.0 with optimal growth occurring at 30–35 (33). The expansive thermo- and halotolerance of *V. alginolyticus* suggests that this bacterium is well adapted to tropical/temperate waters and may only be limited within these systems by seasonal cooling, the presence of freshwater input, and/or atypical hypersaline environments.

Temperature and salinity largely define the broad geographical range of *V. alginolyticus*. However, in warm coastal regions such as Florida and Hawaii, where *V. alginolyticus* infection reports are high (13), other more episodic environmental determinants, such as nutrient availability, may play an important role in shaping the local and short-term *Vibrio* community structure (34–36). Iron is an essential cofactor for bacterial metabolism that is often limiting in marine waters (37, 38). Prior research has established the specific importance of episodic iron input for the enrichment of *Vibrio* populations during Saharan dust deposition events (35). During these events, aerosolized ferric ($Fe^{3+}$) and ferrous ($Fe^{2+}$) iron is transported from Northern Africa via the Atlantic trade winds and deposited into the oligotrophic waters of the Southeastern United States and the Gulf of Mexico (35, 36, 39). Microbial community surveys have shown that these events trigger a substantial increase in the relative and absolute abundance of *Vibrio* in the microbial population, which can swell to 5–30× the background concentration for 24–72 h following the onset of deposition (35, 36, 40). Termed "*Vibrio* blooms," these events have the potential to increase the risk of exposure to opportunistic *Vibrio* pathogens, including *V. alginolyticus*, and are important but understudied factors to consider for risk characterization.

In addition to facilitating population growth, iron acquisition is an important characteristic of virulence for *V. alginolyticus* (41). *V. alginolyticus* has developed a

sophisticated iron acquisition system designed to compete for and scavenge iron from the ambient environment. The two major factors that comprise this system are siderophores and the TonB energy transduction system (41–43). Siderophores are small molecular weight compounds that have a high affinity to chelate ferric iron. These compounds are secreted extracellularly where they bind ambient iron and are recognized by outer membrane proteins (42, 44, 45). Ferrisiderophore complexes are internalized via TonB, a transmembrane protein system that facilitates transfer of energy from the inner cell membrane to the outer cell membrane, enabling active transport (41, 42). While these systems enhance the competitiveness of *V. alginolyticus* in environmental settings, they also contribute to its establishment during infection by outcompeting host iron sequestration mechanisms or directly scavenging iron from heme in blood cells, thus increasing the iron pool available to infecting cells (41, 43). Increased iron availability is known to increase bacterial replication (46) and promote biofilm formation (47) in *Vibrio* spp., which can contribute to the onset and severity of infection. Despite this importance, the relationship between iron concentrations in seawater and *V. alginolyticus* growth and metabolism is poorly understood, and there is a substantial need for baseline characterization.

Here we investigate growth characteristics of *V. alginolyticus* in response to a range of temperature, salinity, and iron concentrations to better understand how these factors can influence population responses across tested strains and to provide context for rapid growth that supports local bloom formation. Additionally, the metabolic response of iron stimulation was further evaluated in a recently isolated environmental strain using gas chromatography-mass spectrometry (GC-MS)-based metabolomic profiling to better understand the specific biochemical response elicited by this bacterium in relation to iron supplementation and deprivation. Together, these findings can be used to better predict the environmental conditions favorable to the proliferation of this bacterium and can be used to mitigate infection risk for humans and in aquaculture settings.

## RESULTS

The results of growth kinetics experiments demonstrated the optimal and tolerable limits of temperature, salinity, and iron concentration for three unique strains of *V. alginolyticus*. Growth curves for all tested strains were constructed from $OD_{600}$ measures to determine the duration of lag phase and the doubling time, which represented the time required to adapt and the productivity of the strain under the given environmental conditions, respectively. Optimal range was defined as the conditions where all three strains demonstrated the fastest strain-level doubling time, whereas the tolerable limit was defined as the conditions where no growth inhibition was observed. Tested strains included two unique environmental isolates, JW16-551 and JW16-580, originally collected in 2016 from water near Looe Key Reef, off the coast of the Florida Keys (USA), during a Saharan dust deposition event (26), and the *V. alginolyticus* type strain, American Type Culture Collection (ATCC) 17749, originally isolated in 1961 from spoiled fish in Japan (48).

### Temperature effects on growth

Optimal *V. alginolyticus* growth occurred between 30°C and 36°C for all strains when grown at a 3% NaCl concentration in non-iron-limiting media [lysogeny broth (LB), with an estimated iron content of 17 µM (49)] (Fig. 1; Fig. S1; Table S4). The fastest doubling time was observed at 32°C (81.6 min), 36°C (71.3 min), and 30°C (96.4 min) for strains JW16-551, JW16-580, and ATCC 17749, respectively. The shortest lag phase duration was observed at 40°C (2.0 h), 40°C (2.2 h), and 36°C (2.7 h) for strains JW16-551, JW16-580, and ATCC 17749, respectively. Within the tested temperature range, all three *V. alginolyticus* strains showed similar patterns of doubling time and lag phase duration up to 36°C. At temperatures ≥36°C doubling time diverged with strain JW16-580, showing a relatively unchanged rate, a progressively longer doubling time for JW16-551, and a substantial increase in doubling time for strain ATCC 17749. A similar divergence

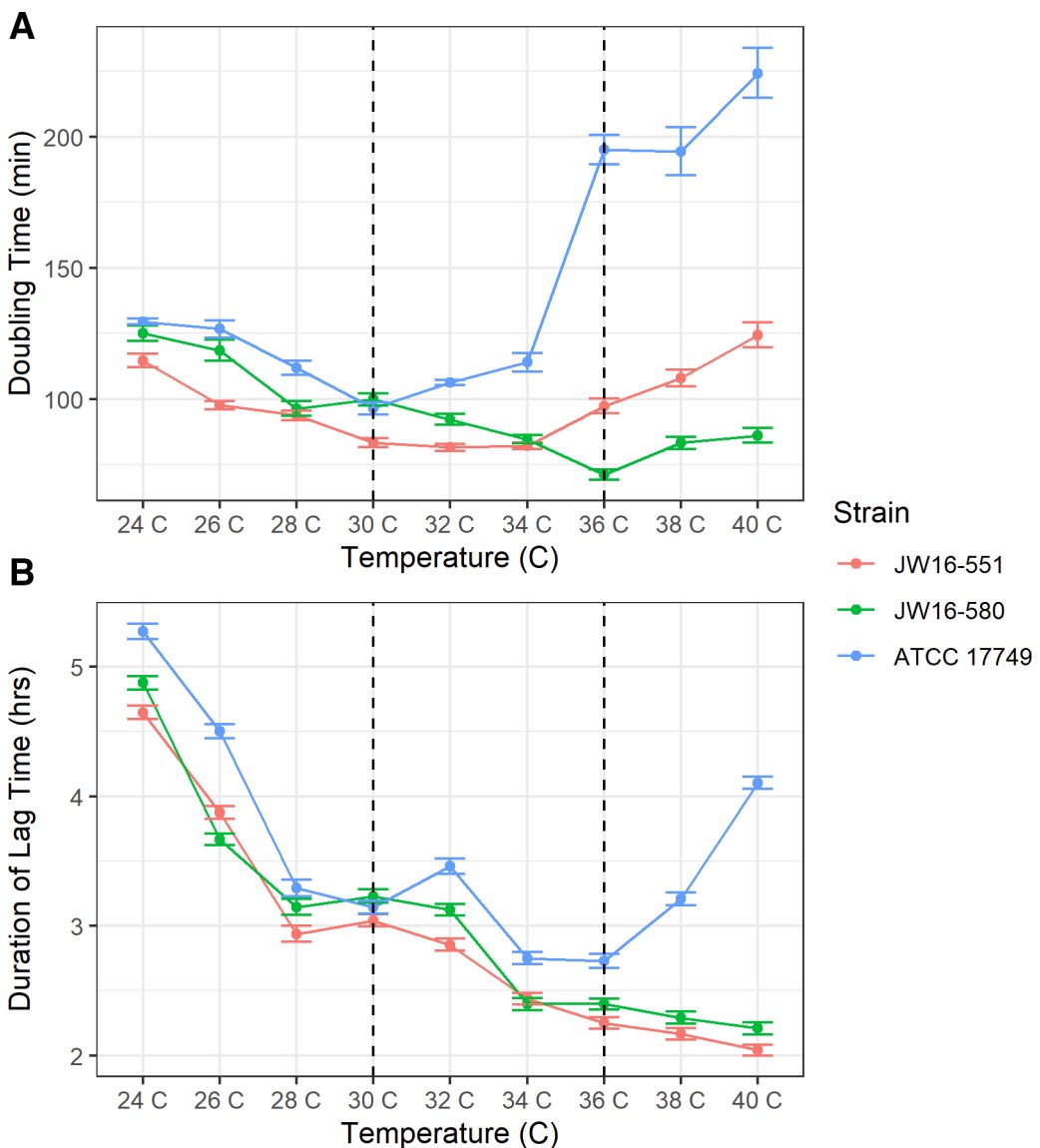

**FIG 1** Growth response of *V. alginolyticus* at varying temperatures. Optimal growth range indicated by dashed vertical lines. All cultures grown in lysogeny broth plus salt (LBS) 3% (wt/vol) NaCl under non-limiting iron conditions. Linerange values represent the standard error of reported metrics. (A) *V. alginolyticus* doubling time from 24°C to 40°C. (B) *V. alginolyticus* lag phase duration from 24°C to 40°C. *N* = 12 for each *V. alginolyticus* strain.

was noted for lag phase duration at temperatures of ≥38°C, where time in lag phase continued to shorten for strains JW16-551 and JW16-580 but increased for strain ATCC 17749 at elevated temperatures. Growth was not inhibited within the tested temperature range (24°C–40°C); however, longer doubling times and lag phase durations were observed at temperatures ≤26°C and ≥38°C for all strains.

## Salinity effects on growth

Optimal *V. alginolyticus* growth occurred between 2% and 4% (wt/vol) NaCl concentrations when cultures were incubated at 30°C in non-iron-limiting media (LB) (Fig. 2; Fig. S2; Table S4). Fastest doubling time was observed at NaCl concentrations of 2% (90.6 min), 3% (91.5 min), and 4% (141.6 min) for strains JW16-551, JW16-580, and ATCC 17749, respectively. The shortest lag phase duration was observed at NaCl concentrations of 2% (2.8 h), 3% (2.5 h), and 3% (3.0 h) for strains JW16-551, JW16-580, and ATCC 17749,

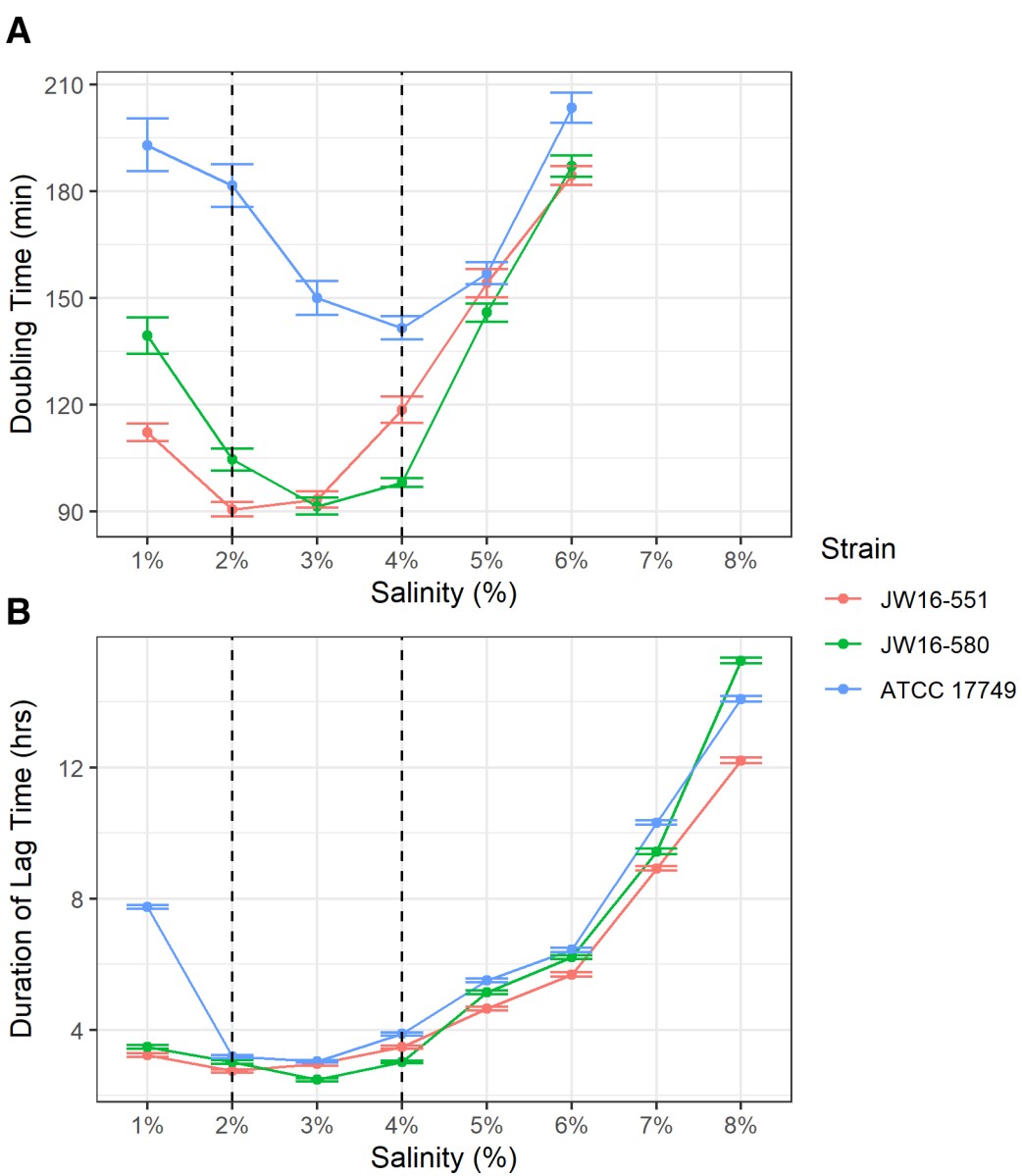

**FIG 2** Growth response of *V. alginolyticus* at varying NaCl concentrations. Optimal growth range indicated by dashed vertical lines. All cultures grown in non-iron-limiting LBS broth amended to the NaCl concentration designated by the experimental condition and incubated at 30°C. Linerange values represent the standard error of reported metrics. (A) *V. alginolyticus* doubling time from 1% to 8% (wt/vol) NaCl. (B) *V. alginolyticus* lag phase duration from 1% to 8% (wt/vol) NaCl. Substantial inhibition of all strains was observed at salt concentrations of ≥7%, preventing accurate calculation of doubling time. However, minor increases in optical density were detected; thus, lag time duration measures were collected for these concentrations. *N* = 12 for each *V. alginolyticus* strain.

respectively. At a NaCl concentration of 1%, doubling time slowed for all tested strains. Lag phase duration remained relatively stable at 1% for both environmental strains but was notably longer for ATCC 17749. Complete inhibition (no growth) was observed in salt-free trials (0% NaCl) for all strains. At increased salinities (5%–8% NaCl), a progressive slowing in doubling time and lengthening of lag phase duration were noted for all strains. Substantial inhibition of growth occurred at NaCl concentrations of ≥7%, which prevented accurate calculation of bacterial doubling time, although sufficient growth was observed to allow determination of the lag phase duration at these concentrations.

## Iron effects on growth

Environmental *V. alginolyticus* strains were amenable to growth at all measured iron concentrations (0.2–20.0 µM as provided in FeCl$_3$) when incubated at 30°C with a 3% (wt/vol) NaCl concentration in defined minimal media (termed VibFeL). However, strain ATCC 17749 was substantially inhibited by the minimal media regardless of iron concentration. This inhibition prevented accurate calculation of doubling time and lag phase duration for most experimental trials with this strain, although detection of minimal growth at iron concentrations of ≥3 µM enabled determination of lag phase duration from 3 to 20 µM and doubling time at 20 µM (Fig. 3; Fig. S3; Table S4). Of the two environmental strains, the fastest doubling time was observed at 20 µM (69.1 min) and 10 µM

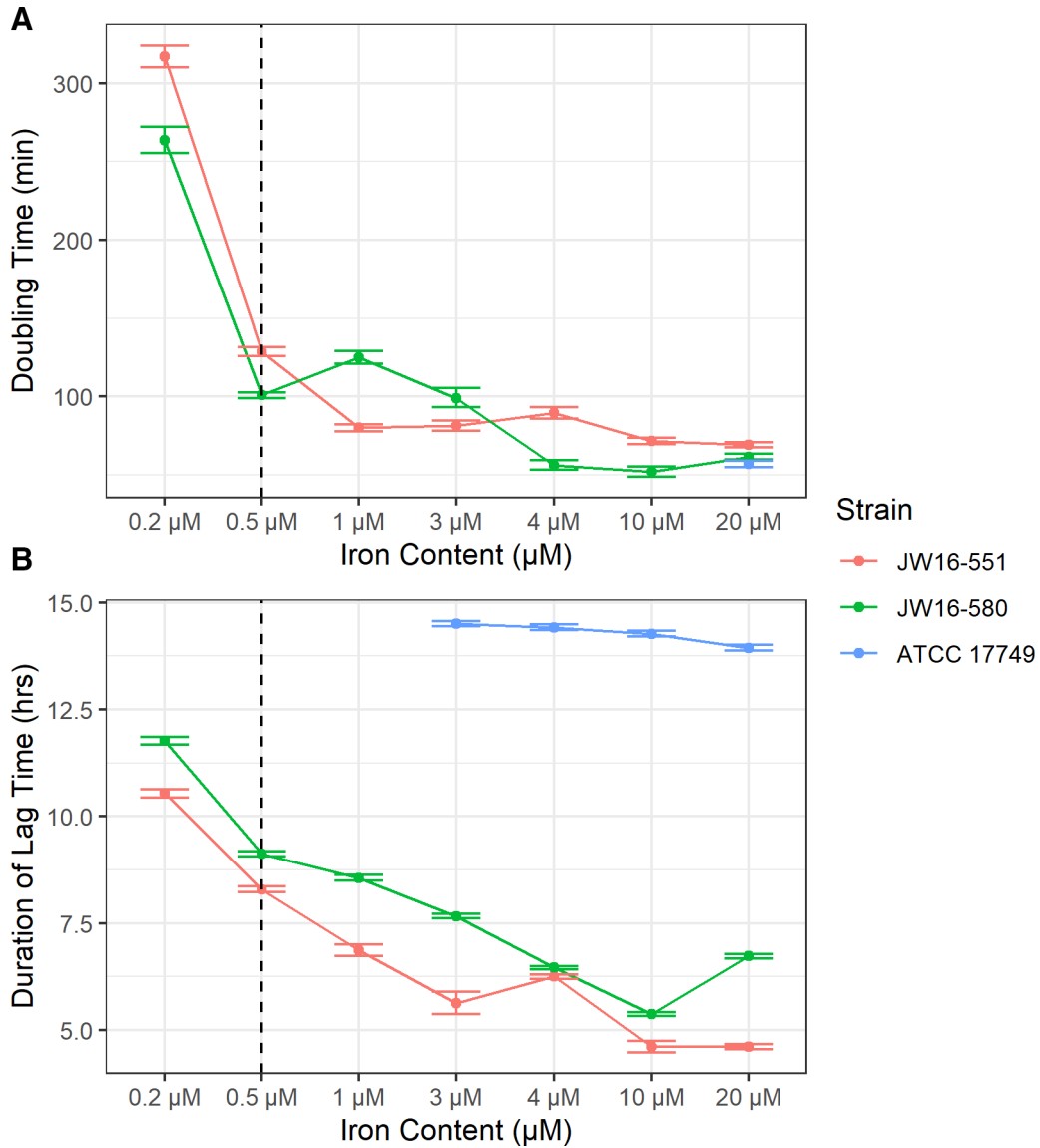

**FIG 3** Growth response of *V. alginolyticus* at varying iron concentrations. Optimal growth occurred at all values ≥0.5 µM (indicated by dashed vertical line) with no discernable upper limit. All cultures grown in VibFeL broth at 3% NaCl (wt/vol) and incubated at 30°C. Linerange values represent the standard error of reported metrics. (A) *V. alginolyticus* doubling time from 0.2- to 20.0-µM iron. (B) *V. alginolyticus* lag phase duration from 0.2- to 20.0-µM iron. Growth of strain ATCC 17749 was substantially inhibited at all tested concentrations of iron; thus, accurate calculation of the growth rate was not possible for this strain except for the 20-µM concentration. Minor increases in optical density were observed at iron concentrations of ≥3 µM, allowing for calculation of lag phase duration from 3 to 30 µM. $N = 12$ for each *V. alginolyticus* strain.

(52.0 min), and the shortest lag phase duration was observed at 10 µM (4.7 and 5.4 h) for strains JW16-551 and JW16-580, respectively. Both environmental strains demonstrated similar patterns of doubling time response throughout the experiment with faster rates observed between 0.5- and 20.0-µM iron concentrations and markedly slowed rates at 0.2 µM. This was also observed for lag phase response where increasing iron facilitated a progressively shorter lag phase duration for both environmental strains, peaking at 10–20 µM. No growth inhibition was observed at an iron concentration of 20 µM; therefore, no upper optimal or tolerable limit could be determined.

## GC-MS metabolomics

Endo- and exometabolite profiles for *V. alginolyticus* strain JW16-551 were compared across four different conditions related to the iron content of the initial culture used for inoculation (referred to as the starvation condition) and the experimental culture (referred to as the iron condition). These trials included (i) non-starved, iron replete (NSFe+), where cultures were initially grown under non-limiting iron conditions and inoculated into iron-replete experimental media (4-µM $FeCl_3$); (ii) non-starved, iron deficient (NSFe-), where cultures were initially grown under non-limiting iron conditions and inoculated into iron-deficient experimental media (0-µM $FeCl_3$); (iii) starved, iron replete (SFe+), where cultures were initially starved of iron for 5 days in iron-deficient media then inoculated into iron-replete experimental media; and (iv) starved, iron deficient (SFe-), where cultures were initially starved of iron for 5 days in iron-deficient media then inoculated into iron-deficient experimental media (Fig. 4). Growth was substantially reduced in all trials using the iron-deficient experimental media regardless of prior starvation condition (Fig. 5). At 18 h of growth, iron-replete cultures (NSFe+ and SFe+) reached a mean of $3.50 \times 10^7$ and $4.03 \times 10^7$ colony forming units (CFU)/mL, whereas, iron-deficient cultures (NSFe- and SFe-) grew to a mean of $2.07 \times 10^6$ and $6.53 \times 10^5$ CFU/mL for non-starved and starved cultures, respectively. This equates to a 15.9-fold

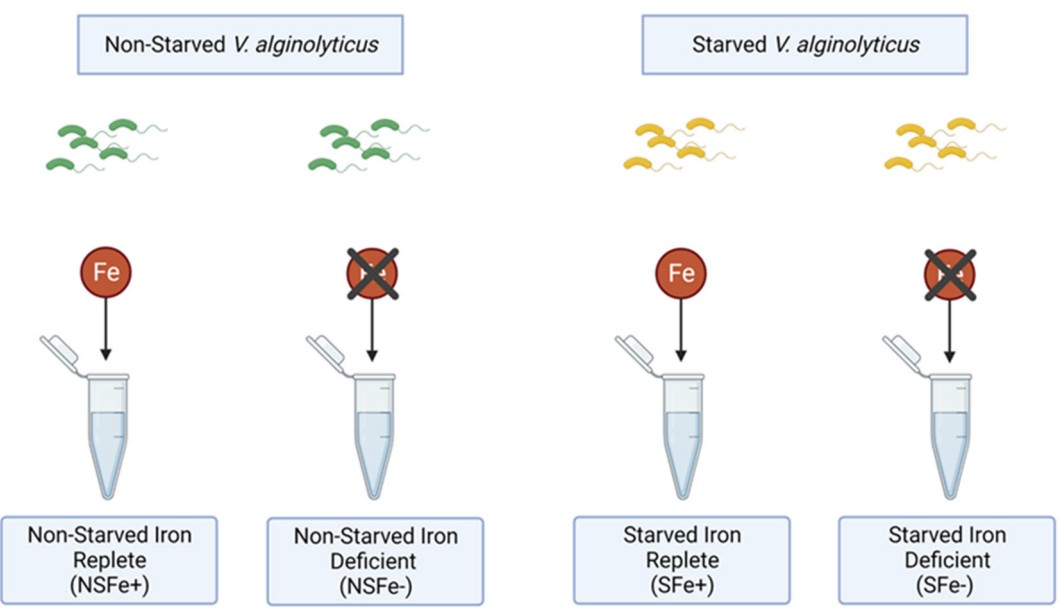

**FIG 4** Sample preparation scheme for iron metabolomics experiments. Starvation condition [non-starved (NS) or starved (S)] represents the iron content of the initial inoculum culture where NS cultures were grown in non-limiting LBS 3% broth for 18 h at 30°C before inoculation, and S cultures were grown in iron-deficient VibFeL (0-µM $FeCl_3$) for 5 days before inoculation. Iron condition (Fe+ or Fe−) represents the iron content of the experimental culture where iron-replete (Fe+) cultures were grown in VibFeL broth amended with 4-µM $FeCl_3$ and iron deficient (Fe−) were grown in non-amended VibFeL broth (0-µM $FeCl_3$). All cultures were inoculated with *V. alginolyticus* strain JW16-551. All experimental VibFeL broth cultures were amended to 3% (wt/vol) NaCl concentration and incubated aerobically for 18 h at 30°C under 100 rpm of shaking agitation.

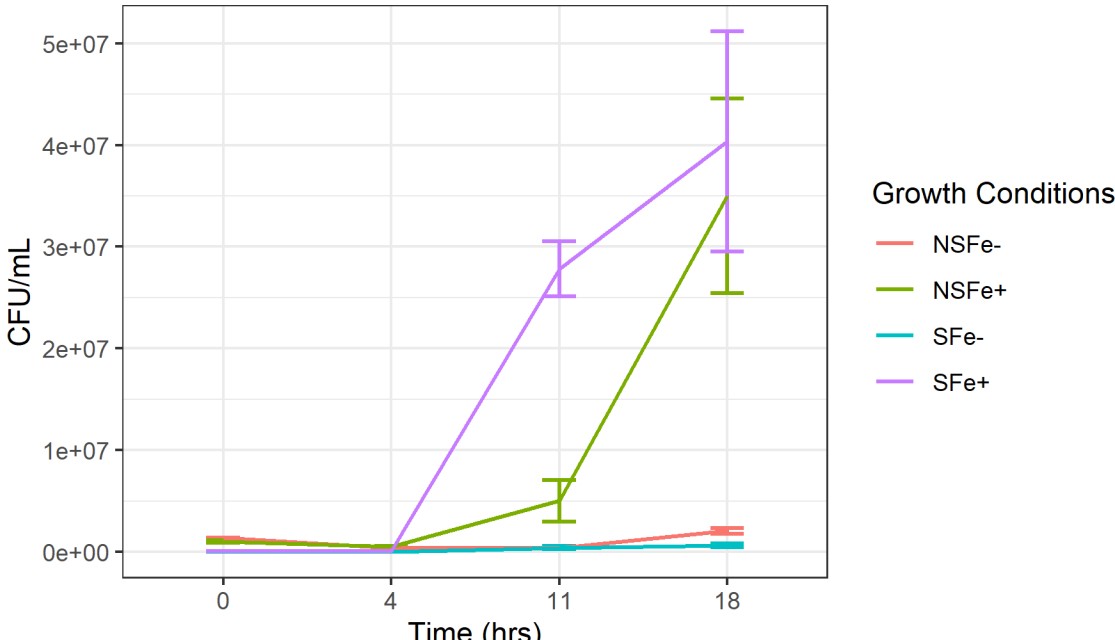

**FIG 5** *V. alginolyticus* growth response (CFU/mL) of iron metabolomic samples. Starvation conditions (NS or S) represent the iron content of the initial inoculation culture, and iron conditions (Fe+ or Fe−) represent the iron content of the experimental culture. NSFe+ represents non-starved iron-replete cultures; NSFe− represents non-starved iron-deficient cultures; SFe+ represents starved iron-replete cultures; and SFe− represents starved iron-deficient cultures. Cultures measured at 0, 4, 11, and 18 h prior to collection for GC-MS analysis.

($P$ value = 0.07) and 60.7-fold ($P$ value = 0.06) increase in culturable *V. alginolyticus* under iron-replete conditions for non-starved and starved cultures, respectively. Furthermore, pre-starved cultures responded more rapidly when transferred to iron-replete media compared to non-starved cultures. Pre-starved cultures showed a mean of $2.78 \times 10^7$ CFU/mL at 11 h of growth in iron-replete media, whereas non-starved cultures only reached $5.03 \times 10^6$ CFU/mL at the same time point, representing a 4.5-fold increase ($P$ value = 0.003) based on starvation (Fig. 5).

## Endometabolites

Cell pellets were extracted to evaluate the endometabolomic response under differing iron and starvation conditions. Principal component analysis (PCA) shows distinct grouping of the cultures by exposure condition (Fig. 6). Fluxes in the endogenous metabolome of iron-replete cultures show similar patterns of clustering and confidence interval overlap regardless of starvation condition, whereas iron-deficient samples show starvation-dependent groupings with minor overlap in principal component space. Comparison of iron conditions (NSFe+ vs NSFe−, and SFE+ vs SFe-) showed increased metabolic activity following transfer to iron-replete media with 49 and 47 significantly elevated metabolites identified in iron-replete trials for non-starved and starved cultures, respectively, compared to 20 elevated metabolites identified from iron-deficient trials (both starvation conditions) (Table 1). Pathway analysis of metabolites from replete cultures (NSFe+ and SFe+) were found to be associated with 25 and 30 unique metabolic pathways (≥2 constituents detected) for non-starved and starved cultures, respectively. Alanine, aspartate, and glutamate metabolism was the most strongly represented pathway (the pathway with the greatest proportion of associated metabolites detected) under both starvation conditions (Fig. 7). Conversely, metabolites from iron-deficient experimental conditions corresponded to only six and four pathways for non-starved and starved cultures, respectively. Aminoacyl-tRNA biosynthesis was the most strongly represented pathway regardless of prior starvation condition (SFe− and NSFe−). No

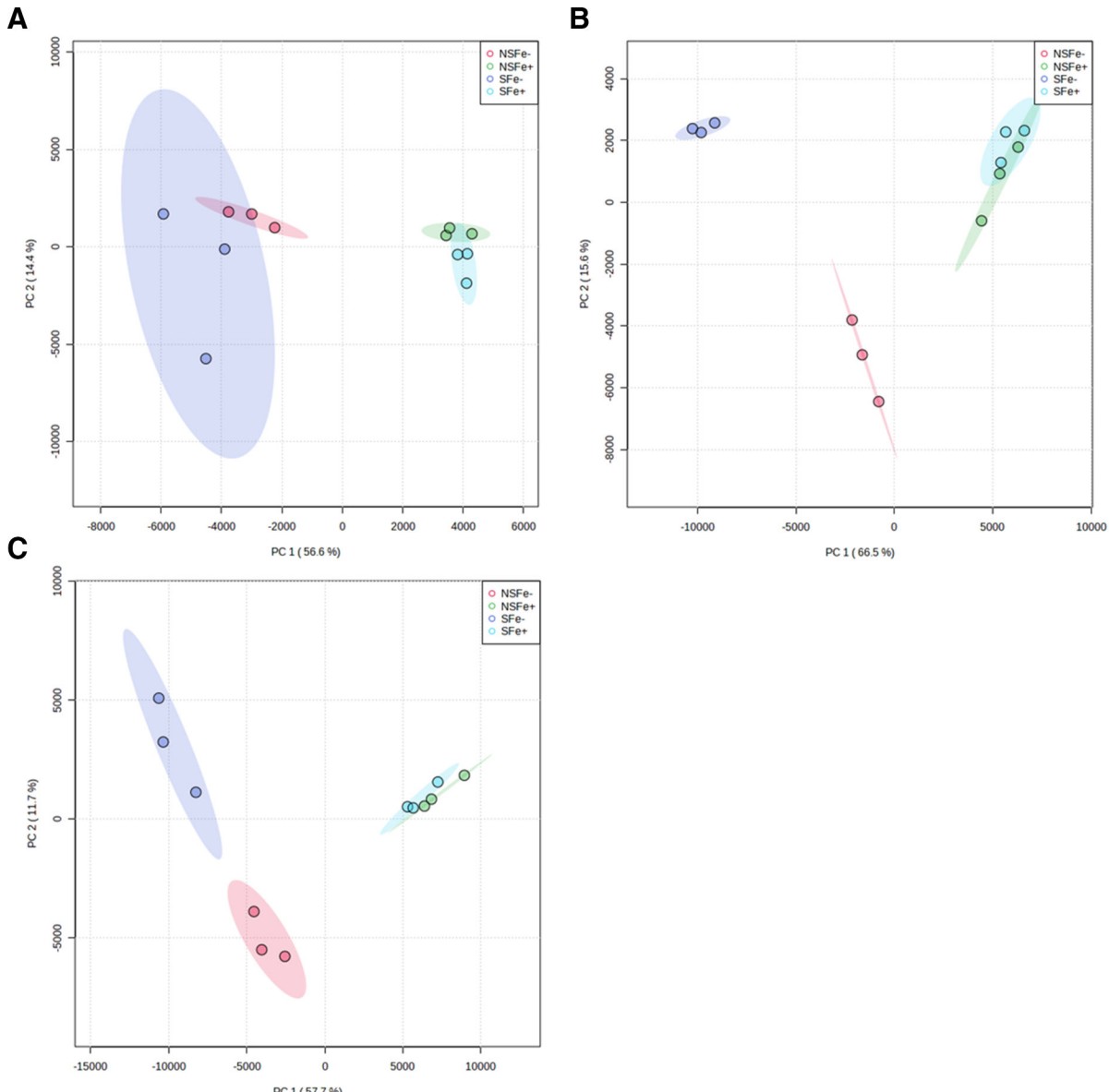

**FIG 6** Principal component analysis of spectral features identified in strain JW16-551 GC-qToF/MS-based metabolomics; (A) polar endometabolites, (B) non-polar endometabolites, and (C) exometabolites. Shaded regions represent a 95% confidence interval of the sample group. $N = 3$ for each sample type.

unique metabolic pathways were detected in iron-deficient samples that were absent in iron-replete samples (Table 1; Fig. 7; Table S1; Fig. S4, S8, and S9).

Comparison of prior growth condition demonstrated that iron starved cultures responded more robustly when transferred to iron-replete media compared to non-starved cultures. Previously starved cultures transferred to iron-replete media (SFe+) showed significant upregulation of 30 unique metabolites corresponding to 19 different metabolic pathways with alanine, aspartate, and glutamate metabolism and beta-alanine metabolism as the most represented pathways. Comparatively, previously non-starved cultures transferred to iron-replete media (NSFe+) showed upregulation of 14 metabolites corresponding to only one biochemical pathway, glycerophospholipid metabolism. Transfer to iron-deficient media demonstrated little difference in the total number of significantly upregulated metabolites and associated pathways with 19 and 12 metabolites corresponding to 3 and 2 metabolic pathways for starved (SFe−) and non-starved (NSFe−) trials, respectively (Table 1; Fig. 7; Table S1; Fig. S5, S10, and S11).

**TABLE 1** Summary of upregulated metabolites and associated metabolic pathways identified for iron and starvation comparisons

| Sample comparison[a,b] | Iron condition | Starvation condition | Metabolite type | Number of upregulated metabolites identified | Number of associated metabolic pathways |
|---|---|---|---|---|---|
| NSFe+/NSFe− | Replete | Non-starved | Endometabolites | 49 | 25 |
| NSFe−/NSFe+ | Deficient | Non-starved | Endometabolites | 20 | 6 |
| SFe+/SFe− | Replete | Starved | Endometabolites | 47 | 30 |
| SFe−/SFe+ | Deficient | Starved | Endometabolites | 20 | 4 |
| NSFe+/SFe+ | Replete | Non-starved | Endometabolites | 14 | 1 |
| SFe+/NSFe+ | Replete | Starved | Endometabolites | 30 | 19 |
| NSFe−/SFe− | Deficient | Non-starved | Endometabolites | 12 | 2 |
| SFe−/NSFe− | Deficient | Starved | Endometabolites | 19 | 3 |
| NSFe+/NSFe− | Replete | Non-starved | Exometabolites | 19 | 14 |
| NSFe−/NSFe+ | Deficient | Non-starved | Exometabolites | 15 | 6 |
| SFe+/SFe− | Replete | Starved | Exometabolites | 30 | 14 |
| SFe−/SFe+ | Deficient | Starved | Exometabolites | 10 | 6 |
| NSFe+/SFe+ | Replete | Non-starved | Exometabolites | 9 | 3 |
| SFe+/NSFe+ | Replete | Starved | Exometabolites | 11 | 9 |
| NSFe−/SFe− | Deficient | Non-starved | Exometabolites | 23 | 10 |
| SFe−/NSFe− | Deficient | Starved | Exometabolites | 9 | 6 |

[a]Sample comparison indicates the two metabolite profiles that were compared where elevated metabolite and pathway totals correspond to the sample in the numerator. Starvation conditions (NS and S) indicate the iron conditions of the initial inoculum culture, whereas iron conditions (Fe+ or Fe−) indicate the iron conditions of the experimental culture.
[b]Non-starved iron replete (NSFe+), non-starved iron deficient (NSFe−), starved iron replete (SFe+), and starved iron deficient (SFe−).

## Exometabolites

Exometabolomic assessments compared the extracellular metabolomic profiles of *V. alginolyticus* spent media in response to prior starvation and iron growth conditions. Results were consistent with those observed from endometabolite analyses, showing similar PCA patterns with respect to iron and prior starvation comparisons. PCA of the detected exometabolites showed distinct clustering by sample type where cultures transferred to iron-replete conditions showed similar patterns of grouping (NSFe+ and SFe+) and iron-deficient cultures separated markedly by prior starvation condition (NSFe− and SFe−) (Fig. 6). Comparison of iron condition suggested an increase in metabolic response in conjunction with transfer to iron-replete media with significant upregulation of 19 and 30 metabolites for non-starved (NSFe+) and starved (SFe+) cultures, respectively (Table 1; Fig. 8). These compounds were mapped to fluxes in 14 total metabolic pathways for both starvation conditions (NSFe+ and SFe+) with alanine, aspartate, and glutamate metabolism representing the most represented pathway. Cultures transferred to iron-deficient media showed significant upregulation of 15 and 10 metabolites associated with six metabolic pathways each with glyoxylate and decarboxylate metabolism and C5-branched dibasic acid metabolism as the most represented for non-starved (NSFe−) and starved (SFe−) cultures, respectively. Of the detected pathways, five (amino sugar and nucleotide sugar metabolism, C5-branched dibasic acid metabolism, galactose metabolism, gluconeogenesis/glycolysis, and methane metabolism) were only identified in iron-deficient cultures (Table 1; Fig. 8; Table S2; Fig. S6, S8, and S9).

Comparison of prior starvation condition was also consistent with endometabolite results with increased metabolic activity in previously starved cultures when transferred to iron-replete conditions. Starved iron replete (SFe+) cultures showed significantly elevated levels of 11 metabolites corresponding to 9 metabolic pathways with alanine, aspartate, and glutamate metabolism identified as the most represented pathway. Comparatively, previously non-starved cultures transferred to iron-replete conditions (NSFe+) showed significant elevation of nine metabolites corresponding to three pathways, alanine, aspartate, and glutamate metabolism, glutathione metabolism, and aminoacyl-tRNA-biosynthesis. Conversely, the number of metabolites and associated

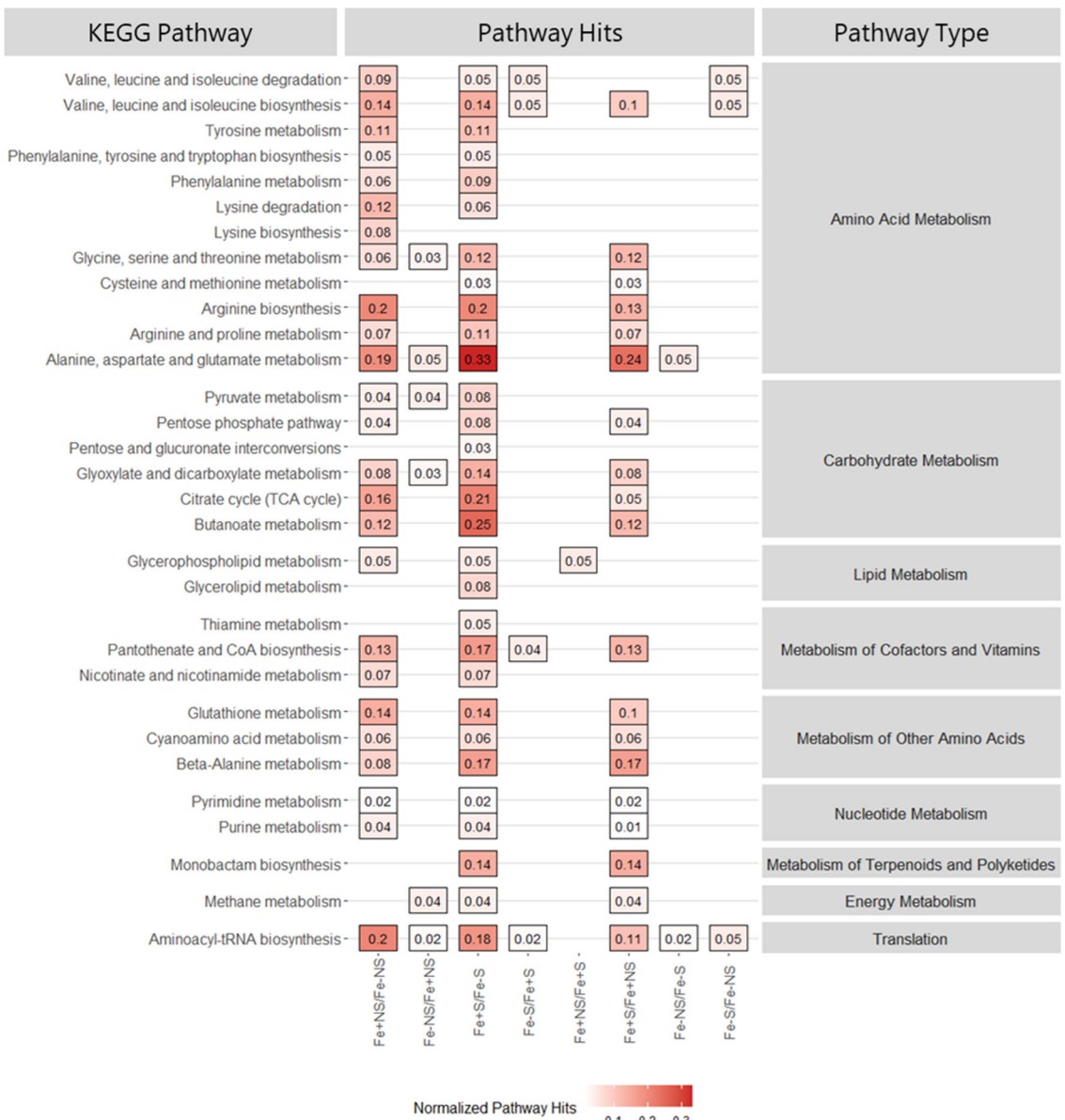

**FIG 7** Metabolic pathways associated with significantly altered endometabolites detected in *V. alginolyticus* cultures under iron supplementation and iron starvation conditions. The left *y*-axis lists all associated Kyoto Encyclopedia of Genes and Genomes (KEGG) pathways; the right *y*-axis illustrates the broad category of each KEGG pathway; the fill color represents the normalized number of pathway hits found for the metabolites detected; and the *x*-axis shows the experimental comparison. From left to right, columns 1–4 illustrate iron comparisons and columns 5–8 represent starvation comparisons. TCA, tricarboxylic acid cycle. CoA, coenzyme A.

metabolic pathways were higher in previously non-starved cultures transferred to iron-deficient media (NSFe−) than in starved iron-deficient cultures (SFe−) with significant upregulation of 9 and 23 metabolites corresponding to 6 and 10 metabolic pathways for starved and non-starved cultures, respectively (Table 1; Fig. 8; Table S2; Fig. S7, S10, and S11).

## DISCUSSION

As a naturally occurring pathogen, exposure risk for *V. alginolyticus* is strongly associated with its population abundance in the environment. Prior research has documented the importance of temperature and salinity as the two major factors influencing *Vibrio* populations, broadly, with temperature, in particular, playing an important role in

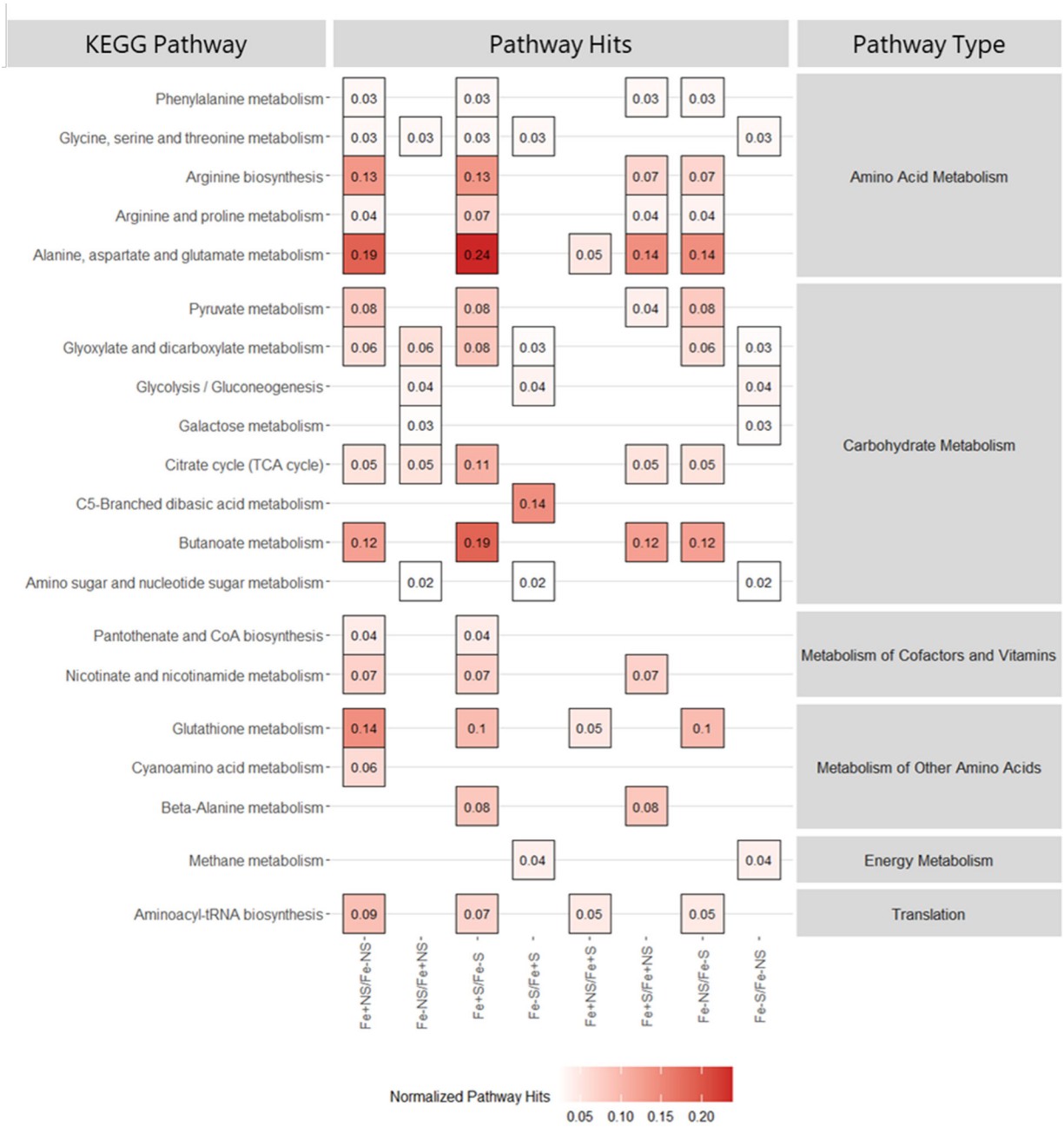

**FIG 8** Metabolic pathways associated with significantly upregulated exometabolites detected in *V. alginolyticus* cultures under iron supplementation and iron starvation conditions. The left *y*-axis lists all associated KEGG pathways; the right *y*-axis illustrates the broad category of each KEGG pathway; the fill color represents the normalized number of pathway hits found in the metabolites detected; and the *x*-axis shows the experimental comparison. From left to right, columns 1–4 illustrate iron comparisons and columns 5–8 represent starvation comparisons.

controlling the geographical range across many *Vibrio* species [e.g., see references (20, 21)]. However, in tropical and subtropical regions, the influence of these factors wanes as ambient conditions rest well within the tolerable limits of most pathogenic *Vibrio* spp. (49). Thus, in these regions, ephemeral *Vibrio* blooms occur following shifts in other environmental conditions, including periodic influx of iron (35, 40, 50). These blooms may increase the risk of infection for opportunistic pathogens such as *V. alginolyticus* and are critical for understanding the population dynamics of this understudied species. Through this research, we described the physiological response of three *V. alginolyticus*

strains in relation to changing temperature, salinity, and iron availability. Furthermore, we demonstrated the importance of iron availability as a key limiting nutrient for the stimulation of *V. alginolyticus* metabolism.

## Growth kinetics

The results of temperature and salinity assessment showed that all tested *V. alginolyticus* strains were amenable to growth at all measured temperatures (24°C–40°C) and NaCl concentrations from 1% to 6% (wt/vol) with optimal growth occurring at 30%–36°C and 2%–4% NaCl in LB. While these values are consistent with previously reported optimal and tolerable limits for this species (30, 33), we note important strain-specific growth variations within these limits. Notably, the two recent environmental strains, JW16-551 and JW16-580, demonstrated increased thermo- and halotolerance compared to the ATCC type strain 17749. These differences were most evident at temperatures ≥34°C and NaCl concentrations ≤3% where the doubling time and/or lag phase duration for ATCC 17749 increased substantially compared to the other tested strains. While the exact mechanism for this difference is unknown within the scope of this study, we hypothesize it may be due to differences in the time in culture from the collection of the ATCC strain compared to the environmental strains and/or horizontally acquired adaptations due to the specific environment where the isolate was obtained (51–53).

The results of iron growth kinetics experiments differed from temperature and salinity evaluations in that a distinctive tolerance range and optimal iron concentration was not observed. All iron amendments ≥0.5-µM $FeCl_3$ enabled growth of environmental *V. alginolyticus* strains (JW16-551 and JW16-580) at 30°C and 3% NaCl in VibFeL minimal media, with a dramatic reduction in doubling time and lag phase duration between 0.2- and and 0.5-µM amendments. For JW16-551, doubling time plateaued starting around 1-µM $FeCl_3$ amendment but not until 4-µM $FeCl_3$ for JW16-580. Duration in lag phase continued to progressively decline before reaching a plateau of 4.5–5.5 h at ~10 µM for both strains. The ATCC type strain, which was originally isolated from spoiled fish in 1961, only grew under the highest iron amendments in this study. Marine waters are highly iron limited with typically <1.0 nM of dissolved Fe (38, 54). The significant reduction in lag and doubling time for environmental *V. alginolyticus* strains with Fe amendments between 0.2 and 0.5 µM, confirms prior observations that episodic iron input (e.g., from aeolian sources) may facilitate rapid population expansion (i.e., blooms). This is consistent with the observations of Westrich et al. (35), where *V. alginolyticus* levels increased substantially following the addition of simulated Saharan dust at iron levels ranging from 0 to 0.84 µM. This finding is particularly important for oligotrophic systems such as coral reefs which typically possess lower iron pools (55, 56) compared to estuarine systems (38, 57) and may be more prone to the formation of iron-induced *Vibrio* blooms.

Under the scope of the present research, it is unclear why strain ATCC 17749 was not amenable to growth in the iron-limiting media (VibFeL) regardless of iron concentration. We suspect that this difference may be the result of adaptations associated with geographical and/or isolation source differences between isolation from spoiled fish ATCC 17749 (Japan) and isolation from seawater (JW16-551 and JW16-580; United States, Florida). Additionally, time in culture, 1961 for ATCC 17749 vs 2016 for the JW strains, may have affected growth patterns. Prior research by Westrich et al. (35) successfully utilized VibFeL as an iron-limiting minimal media for the growth of a seawater-derived ATCC *V. alginolyticus* strain 33839; however, this research represents the first attempt to utilize this media with an animal-derived strain. Other work on developing a differential selective *V. alginolyticus* media has stressed the importance of sucrose concentration for the successful growth of *V. alginolyticus* strains (58). It is possible that the relatively low sucrose concentration utilized in VibFeL [0.4% (wt/vol) compared to 2.0% (wt/vol) in thiosulfate bile salts sucrose (TCBS) agar] is limiting the growth of ATCC 17749. Furthermore, due to the unimpeded growth of ATCC 17749 in non-limiting media [LB broth

amended to 3% (wt/vol) NaCl], it is possible that the minimal VibFeL media lacks one or more critical substrates for growth of this strain.

## Iron metabolomics

Metabolomics assessment provides biochemical context for the observed changes in growth kinetics and cell counts upon transfer to iron-replete and iron-deficient conditions. Clustering of both endo- and exometabolites corresponded with prior starvation and growth conditions, with overlap observed when transferred to iron-replete media, suggesting that iron supplementation facilitates activation of similar metabolic pathways regardless of prior starvation condition. Conversely, when transferred to iron-deficient media, cultures showed little to no overlap in component space, suggesting that these treatments are metabolically distinct because of prior starvation. We suspect that this difference may in part be due to the utilization of stored iron (59, 60) by non-starved cultures, facilitating low-level metabolic activity under iron-deficient conditions. This hypothesis is supported by growth observations of these cultures where non-starved cultures showed slightly elevated CFU/mL counts compared to starved cultures after 18 h of growth in iron-deficient media (Fig. 5).

Given that it is an essential trace metal, iron supplementation facilitated an increase in the total number of upregulated metabolites and corresponding metabolic pathways regardless of prior starvation condition for both endo- and exometabolites. Of the enriched pathways detected, those associated with amino acid metabolism were the most impacted, with significant upregulation of metabolic intermediaries and end products linked to amino acid biosynthesis and/or degradation such as succinate, fumarate, L-aspartate, L-alanine, putrescine, γ-aminobutyric acid, glutamate, and 2-oxoglutarate (α-ketoglutarate) after transfer to iron-replete conditions. The availability of iron facilitates increased protein synthesis and is consistent with the established role of iron as a critical cofactor for enzyme catalyzation (61, 62).

Second to amino acid metabolism, carbohydrate metabolism was also highly upregulated when transferred to iron-replete media. Enriched pathways were associated with energy production processes, namely, the tricarboxylic acid cycle (TCA), glycolysis, butonate metabolism, and glyoxylate and decarboxylate metabolism, suggesting that iron is essential for energy generation in *V. alginolyticus*. This finding is consistent with the established iron requirement of *Vibrio* spp. to stimulate replication (63) and provides justification for the substantially reduced level of growth observed in iron-deficient cultures (Fig. 5). Furthermore, the upregulation of glyoxylate and decarboxylate metabolism is of particular interest. Prior research has suggested that the use of the glyoxylate shunt (an anabolic variation of the TCA cycle) may represent an effort to reduce internal iron quota through a reduction in the use of iron-dependent enzymes for energy production (64). *Vibrio* utilizing this mechanism may explain the enrichment of this pathway in both iron-replete and iron-deficient conditions.

To a lesser extent, iron supplementation also enriched pathways associated with lipid, nucleotide, vitamin/cofactor, and secondary (terpenoids and polyketides) metabolism. Like amino acid metabolism, enrichment of these pathways was associated with the upregulation of intermediary and end-point metabolites related with these processes including L-aspartate, L-valine, L-tyrosine, uracil, succinate, thymine, D-ribose 5′-phosphate, adenine, and urea, further demonstrating a broad activation of metabolic processes under replete conditions. It should be noted that iron supplementation also facilitated an enrichment of the translation pathway, aminoacyl-tRNA biosynthesis. While this pathway does not exclusively represent metabolism, greater pathway representation was observed in iron-replete trials, suggesting increased translation activity in response to iron availability. Furthermore, enrichment of methane metabolism was detected in several metabolite comparisons. While interesting to note, enrichment of this pathway in all instances was due to the upregulation of glycine and pyruvate, two metabolites commonly associated with amino acid and carbohydrate metabolism. Thus, we suspect that detection of this pathway is likely the result of false discovery of intermediaries.

No pathways unique to iron-deficient cultures were detected from endometabolite data, suggesting that in the absence of iron *V. alginolyticus* metabolism is largely inhibited. However, exometabolite data demonstrated an impact in three carbohydrate-centric metabolic pathways not found in iron-replete samples: galactose, C5-branched dibasic acid, and amino sugar/nucleotide sugar metabolism. Due to the absence of these pathways in the endometabolite data, we hypothesize that these may be due to detection of residual media carbohydrates (i.e., sucrose) that were not utilized by *V. alginolyticus* in the absence of iron.

Initial iron deprivation (prior to the start of the experiment) facilitated increased metabolic activity once reintroduced to an iron-rich environment. Under iron-replete conditions, prior starved samples showed elevated levels of upregulated endo- and exometabolites corresponding to increased enrichment of amino acid, carbohydrate, vitamin/cofactor, nucleotide, and secondary metabolism pathways. This rapid response is consistent with the observed growth patterns in iron-replete cultures where prior starved samples showed increased CFU per milliliter levels at earlier timepoints (4 and 11 h) compared to previously non-starved cultures, suggesting faster response from these strains (Fig. 5). This is important to note as natural populations of *V. alginolyticus* are expected to be consistently iron deprived in the marine environment (63); thus, this response may indicate how these populations react following iron influx. Additionally, prior starvation did not appear to activate alternate metabolic mechanisms but rather stimulated a more robust or exacerbated response in the identified pathways. Analysis of iron-deficient samples showed variable metabolic results based on starvation, suggesting that in the absence of abundant iron, pre-starvation has little metabolic effect on *V. alginolyticus*.

Beyond pathway enrichment, analysis of upregulated metabolites provided evidence of *V. alginolyticus* iron acquisition mechanisms. Prior research has demonstrated that the *Vibrio*-derived siderophore vibrioferrin is comprised of equal parts L-alanine, citric acid, 2-oxoglutatic acid, and ethanolamine (65–67). In the present study, each of these metabolites were found to be significantly upregulated in analysis of endometabolites, exometabolites, or both under iron-replete conditions. This upregulation suggests that when supplemented with iron, *V. alginolyticus* strain JW16-551 likely produces vibrioferrin or a homologous siderophore as a mechanism of iron acquisition. This finding is consistent with that of Wang et al. (68), who found similar evidence of *V. alginolyticus* production of a vibrioferrin-like siderophore through characterization of the *fur* gene cluster [a known regulator of iron acquisition mechanisms in *Vibrio* spp. (42)] and siderophore purification from low-iron cultures. It should be noted that while detection of these metabolites together suggests the presence of vibrioferrin, this suite of metabolites have functions in other metabolic pathways such as amino acid and lipid metabolism; thus, continued investigation of *V. alginolyticus* siderophore production is needed to corroborate these findings.

## Conclusion

As an indigenous microorganism and opportunistic pathogen, the risk of *V. alginolyticus* infection is directly related to the abundance of its populations in the environment. Prior research has successfully demonstrated the importance of temperature and salinity as critical factors restricting *V. alginolyticus* range and growth. However, few studies to date have examined the importance of tertiary environmental determinants which play an important role in regions where temperature and salinity are non-limiting and where localized conditions may give rise to short-term blooms. Here we reconfirm the broad temperature and salinity tolerance of *V. alginolyticus* and demonstrate the critical importance of iron availability to simulate the growth and metabolism of *V. alginolyticus*. The results of this research provide important context for the environmental response of *V. alginolyticus* populations in relation to iron availability and stresses the importance

of consideration of episodic iron deposition for prediction of *V. alginolyticus* infection risk.

## MATERIALS AND METHODS

### Strains and storage

Experimental *V. alginolyticus* strains were obtained from our culture collection (E.K. Lipp, University of Georgia). Strains consisted of two environmental isolates collected from pelagic waters near Looe Key Reef, FL, during a Saharan Dust event in 2016 (strains JW16-551 and JW16-580) as well as the ATCC strain for *V. alginolyticus* (17749) originally isolated from spoiled fish in Japan in 1961 (Table 2). Physiological evaluation measured the growth response of all three strains at varying temperatures, salinities, and iron content. Metabolomic analysis specifically focused on the iron response of strain JW16-551, which was previously shown to be highly responsive to Saharan dust-derived input of biologically available iron. All parent cultures were stored at −80°C in 20% glycerol (vol/vol, final concentration). Prior to the start of experimentation, strains were revived in 4 mL of LB (Sigma-Aldrich, Miller formulation) amended to 3% wt/vol NaCl (termed LBS 3% henceforth) at 30°C with 100 rpm shaking agitation (New Brunswick Scientific, C24 Incubator Shaker).

### Physiological evaluation

Physiological evaluation compared the growth kinetics of *V. alginolyticus* under changing conditions of temperature, salinity, and iron content. Temperature effects were evaluated from 24°C to 40°C at 2°C intervals controlled by incubation. Salinity effects were measured using NaCl concentration from 0% to 8% (wt/vol) at 1% intervals. NaCl concentration was controlled using "home-brew" LBS media consisting of 10 g of peptone (Sigma-Aldrich), 5 g of yeast extract (Sigma-Aldrich), and NaCl (Sigma-Aldrich) added to the concentration of the desired salinity percentage [NaCl level is designated as the percent value of LBS (i.e., 6% NaCl media is abbreviated as LBS 6% in-text)]. Prior work indicates that LB provides ~17-µM Fe (69). Iron effects were measured between 0.2- and 20.0-µM Fe(III) at 0.2-, 0.5-, 1.0-, 3.0-, 4.0-, 10.0-, and 20.0-µM added $FeCl_3$. These levels were selected to represent a range of iron concentrations from environmentally relevant (0.2–4.0 µM) to highly elevated levels (10–20 µM). Iron concentrations above 20 µM were not evaluated due to the formation of precipitate, which interfered with the accurate collection of optical density measures. Iron concentration was controlled using a custom low-iron media termed VibFeL prepared using the methods of Westrich et al. (35). During VibFeL preparation, ambient iron from the basal media components was removed by chelation through a chromatography column containing Chelex 100 (Sigma-Aldrich) ion-exchange resin. Following removal, iron was restored to the media to the designated experimental concentration through the addition of ferric chloride ($FeCl_3$, Sigma-Aldrich). It should be noted that concentrations are expressed as added amounts of $FeCl_3$. While efforts were taken to minimize any ambient Fe, low background levels may have been present.

To begin growth kinetics experiments, cultures were revived from −80°C storage as described above and incubated overnight (~18 h) to reach stationary phase. Of a $10^{-1}$ dilution of each strain (~$8.1 \times 10^4$, $9.3 \times 10^4$, and $4.7 \times 10^4$ CFU for JW16-551, JW16-580,

**TABLE 2** *V. alginolyticus* strain details and documentation[c]

| Species | Strain designation[a] | Strain type | Isolation source | Citation |
|---|---|---|---|---|
| *V. alginolyticus* | ATCC 17749 | Type strain | Spoiled horse mackerel, Japan | (48) |
| *V. alginolyticus* | JW16-551 | Environmental isolate | Seawater, Looe Key, FL | (36)[b] |
| *V. alginolyticus* | JW16-580 | Environmental isolate | Seawater, Looe Key, FL | (36)[b] |

[a]ATCC strains obtained from the American Type Culture Collection.
[b]Isolates were collected during this study but are not described.
[c]Strains ATCC 17749, JW16-551, and JW16-580 were used for growth kinetics experiments, and strain JW16-551 was used for iron metabolomics experiments.

and ATCC 17749, respectively), 1.5 µL was inoculated into 148.5 µL of the designated media type in a clear 96-well microplate (Nunc Pinch-bar MicroWell 96-Well Microplate, ThermoFisher). Inoculated plates were loaded into a Varioskan LUX microplate reader (ThermoFisher), and growth was evaluated using optical density (OD). OD measures were taken at 600 nm every 150 s for a period of 15 h. All plates were incubated with 120 rpm of continuous shaking agitation in 12 replicates ($N = 12$) for each strain under each growth condition. Unless designated as the experimental variable, plates were incubated at 30°C with 3% (wt/vol) salt content, and a non-limiting supply of biologically available iron (non-chelated media). These values were selected to represent the most optimal environmentally relevant temperature and salinity for *V. alginolyticus* growth. Growth data were analyzed in Rstudio using the packages "tidyverse," "readxl," "SciViews," and "FSA." The duration of lag phase was calculated as the elapsed time required to reach a detectable OD threshold (signal above background noise). This threshold was calculated as the mean of all measurements recorded between an $OD_{600}$ of 0.05 and 0.15 to account for measurement variation. Doubling time was calculated using the standard two-step OD formula:

$$\text{Growth Rate Constant} = \frac{[\ln(\text{LateLogOD}) \ - \ \ln(\text{EarlyLogOD})]}{\text{Time}_{\text{late}} \ - \ \text{Time}_{\text{early}}}$$

$$\text{Doubling Time} = \frac{\ln(2)}{\text{Growth Rate Constant}}$$

Using this equation, LateLogOD represents the OD of the culture toward the end of log phase (~3rd quartile); EarlyLogOD represents the OD at the beginning of log phase (~1st quartile); $\text{Time}_{\text{late}}$ represents the elapsed time to reach the LateLogOD; and $\text{Time}_{\text{early}}$ represents the elapsed time to reach EarlyLogOD. To account for sample selection variation, LateLogOD, EarlyLogOD, $\text{Time}_{\text{late}}$, and $\text{Time}_{\text{early}}$ were calculated as aggregate values within specified OD ranges. This was done to improve representation of these key metrics by including data from multiple close datapoints rather than selection of a single representative datapoint. LateLogOD and $\text{Time}_{\text{late}}$ were calculated as the mean values of OD and elapsed time for all measures ranging from an $OD_{600}$ of 0.65–0.75 for temperature and salinity trials and 0.15–0.20 for iron trials. A reduced range was selected for iron trials due to the overall reduced growth capacity of *V. alginolyticus* under the limiting conditions of VibFeL media. Similarly, EarlyLogOD and $\text{Time}_{\text{early}}$ were calculated as the mean values of OD and elapsed time for all measures ranging from an $OD_{600}$ of 0.10 to 0.20 for temperature and salinity trials and from 0.05 to 0.10 for iron trials. Strain-level doubling times and lag phase durations were tested for significance across all abiotic metrics using a Kruskal-Wallis test and Shapiro-Wilk test for normality (Table S3). Pairwise strain-level comparisons were tested using Dunn's multiple comparisons test with Holm's *P* value adjustment to identify significant differences in the strain-level growth response across treatments (Table S4).

## Iron metabolomic culture conditions

Metabolomic experiments were conducted to explore the biochemical effects of iron availability on *V. alginolyticu*s. These experiments focused on the response of strain JW16-551,which was collected as part of a prior study on Saharan dust deposition in the Florida Keys (36). Cultures were prepared to measure the effects of iron condition and iron starvation. Iron condition experiments compared differences between cultures grown in iron-replete (VibFeL amended with 4-µM $FeCl_3$) and iron-deficient (VibFeL non-amended, ~0-µM Fe) media. Four micromolar was designated as iron replete based on growth experiments for JW16-551 and prior work by Westrich et al. (35). This level is also consistent with iron concentrations found in non-oligotrophic coastal waters (70), offering an environmentally relevant concentration that would be sufficient for growth. Iron-deficient cultures are noted as ~0-µM iron amendments, due to undefined levels of ambient iron contamination from the laboratory space. To reduce the level of contamination, all iron-deficient VibFeL were prepared immediately before use in experimentation

with acid-washed glassware and stored for no more than 4 h. Iron starvation experiments compared the differences between cultures that were initially "starved" of iron for 5 days in iron-deficient media (~0-µM VibFeL) at 30°C and "non-starved" cultures grown for 18 h in non-iron-limiting media (LBS 3%) at 30°C. Non-starved and starved parent cultures were subsequently inoculated into experimental media (either iron replete or iron deficient) for growth and metabolomic measurement.

To prepare experimental cultures, strain JW16-551 was revived from −80°C storage as described above and incubated overnight (non-starved) or for 5 days (starved). One milliliter of cultured cells was removed, pelleted by centrifugation at ~4,000 × $g$ for 2 min, and resuspended in 1 mL of sterile 1× phosphate-buffered saline in triplicate to wash cells of residual media. One hundred microliters of washed cells ($5.00 \times 10^6$ and $1.02 \times 10^4$ CFU for non-starved and starved cultures, respectively) were inoculated into 10 mL of VibFeL media amended to iron-deficient or iron-replete conditions designated by the experimental trial. Inoculated cultures were incubated aerobically for 18 h at 30°C with 100 rpm of shaking agitation (New Brunswick Scientific, C24 Incubator Shaker). At 4, 8, 11, and 18 h, cellular growth was quantified using culture-based plate counts where 100 µL of culture was removed, serial diluted (10-fold), and spread plated with glass rattler beads (Zymo Rattler Plating Beads, 4.5 mm) onto TCBS agar. At 18 h, cultures were removed and pelleted by centrifugation at ~4,000 × $g$ for 10 min, and the supernatant (henceforth termed "spent media") was removed. Cell pellets were immediately quenched in ice-cold 100% methanol (Sigma-Aldrich), transferred to 1.5-mL microcentrifuge tubes, and stored at −20°C for endometabolite analysis. Spent media (1.9 mL) was transferred to a 2-mL microcentrifuge and quenched through the addition of 100 µL of acetone (Sigma-Aldrich) and stored for exometabolite analysis at −20°C.

## Extraction

Prior to analysis, all endometabolite samples were lysed and extracted to target polar and non-polar metabolites using liquid-liquid extraction (71). Samples were dried using a SpeedVac Plus (Savant) for 18 h to remove residual methanol. Dried cell pellets were resuspended in 485 µL of 82.5% methanol:water, and a 3.2-mm diameter stainless steel disruption bead (BioSpec Products Inc.) was added to each sample. Samples were lysed using a Qiagen TissueLyser II bead mill following a step-wise extraction protocol. First, samples were processed for 10 min at a frequency of 15/s. Next, samples were centrifuged for 15 s using a bench top microcentrifuge; 300 µL of chloroform (Sigma-Aldrich) was added; and they were disrupted on the bead mill for 20 min at a frequency of 15/s. Lastly, samples were centrifuged for 15 s using a bench top microcentrifuge; 200 µL of chloroform (Sigma-Aldrich) and 200 µL of dH$_2$O (18.2 MΩ water) were added and returned to the bead mill once more for 10 min at a frequency of 15/s. Following lysis, samples were centrifuged at 1,000 × $g$ for 15 min at 4°C. Centrifugation resulted in the production of two phases: an upper methanol-water phase containing polar metabolites and a lower chloroform phase containing non-polar metabolites separated by a thin layer of protein debris. Each phase was removed and dispensed into a 2-mL glass vial. Care was taken not to disturb the protein debris layer when removing each phase.

Exometabolite (spent media) samples did not require extraction. All samples were retrieved from −20°C storage and thawed at room temperature. Thawed samples were vortexed for 30 s to homogenize the mixture, and 200 µL of spent media was transferred to a 2-mL vial. Both endometabolite (polar and non-polar) and exometabolite samples were dried overnight as described above prior to derivatization.

## Derivatization

Lyophilized samples were derivatized sequentially with methoxyamine hydrochloride (MeOX) (Sigma-Aldrich) and N,O-bis(trimethylsilyl)trifluoroacetamide containing 10% trimethylchlorosilane (BSTFA + 10% TMCS) (Thermo Scientific). For methoxyamination, 60 mg of MeOX was dissolved into 3 mL of pyridine (ThermoFisher), and 30 µL was

added to each sample vial and vortexed for 10 s. All samples were incubated at 60°C for 2.5 h with intermediate vortexing (i.e, every 30 min). After 2.5 h, samples were removed and allowed to cool for 10 min. Fifty microliters of BSTFA was added to each sample and vortexed for 10 s. Sample vials were incubated at 60°C for 1.5 h and removed every 30 min for vortexing. This process was repeated for both endometabolite and exometabolite samples.

## GC-MS analysis

Metabolomics samples were analyzed on an Agilent 8890 gas chromatograph coupled to a 7,250 quadrupole time of flight mass spectrometer (GC/q-ToF-MS) equipped with a DB-5MS ultra inert column (30 m × 250 µm × 0.25 µm; Agilent Technologies) using electron impact ionization scanning from 50 to 600 $m/z$. Samples (1 µL) were injected in split mode at 10:1, and helium was used as the carrier gas. Initial oven temperature was held at 60°C for 1 min then ramped 10°C/min to 325°C and held for 10 min (total runtime 37.5 min). Post-acquisition, spectra were imported into MetAlign (72) for pre-processing and alignment. Vendor-recommended parameters for high-resolution GC/qToF-MS were used. Retention time and $m/z$ paired data:$m/z$ were analyzed using MetaboAnalyst (for PCA analysis) and Rstudio for additional statistical analyses using the R libraries "tidyverse" and "readxl." Retention times were compared by iron and starvation condition resulting in eight major comparisons: not-starved iron-replete vs not-starved iron-deficient (NSFe+/NSFe−), starved iron-replete vs starved iron-deficient (SFe+/SFe−), not-starved iron-replete vs starved iron-replete (NSFe+/SFe+), not-starved iron-deficient vs starved iron-deficient (NSFe−/SFe−), and the inverse of these. Relative concentrations were compared using Student's $t$-test to identify significantly ($P \leq 0.05$) perturbed spectral features from each comparison. Following metabolite identification using both the National Institute of Standards and Technology (NIST) and Agilent's Fiehn Metabolomics libraries, functional analysis of significant metabolites was performed using MetaboAnalyst's "Pathway Analysis" feature with the *Escherichia coli* K-12 MG1655 prokaryote pathway library. To account for differences in the total number of metabolites per pathway, pathway hits were normalized using the equation

$$zi = \frac{(x_i - \min(x))}{(\max(x) - \min(x))}$$

where $z_i$ is the normalized value; $x_i$ is the total pathway hits; min($x_i$) is the minimum pathway hits, or 1; and $max(x)$ is the maximum pathway hits or the total metabolites in the pathway. All pathways with a hit count of 1 were removed from analysis to account for the possibility of false discovery.

## ACKNOWLEDGMENTS

We thank the U.S. Environmental Protection Agency (EPA), Office of Research and Development, and Center for Environmental Measurement and Modeling for their assistance with metabolomics processing and instrumentation. We thank the University of Georgia, Graduate School, for their financial support of this project through the 2022 Summer Research Grant program. We also thank Ms. Rachel Phan, Ms. Carolina Melendez-Declet, and Mr. Carter Coleman for their assistance with laboratory processing.

The views expressed in this article are those of the authors and do not necessarily represent the views or policies of the U.S. Environmental Protection Agency. Any mention of trade names or commercial products does not constitute EPA endorsement or recommendation for use.

## AUTHOR AFFILIATIONS

[1]Department of Environmental Health Science, University of Georgia, Athens, Georgia, USA

[2]U.S. Environmental Protection Agency, Office of Research and Development, Center for Environmental Measurement and Modeling, Athens, Georgia, USA

## AUTHOR ORCIDs

William A. Norfolk ⬥ http://orcid.org/0000-0002-3927-0361
Erin K. Lipp ⬥ http://orcid.org/0000-0002-8066-0636

## AUTHOR CONTRIBUTIONS

William A. Norfolk, Conceptualization, Data curation, Formal analysis, Investigation, Methodology, Resources, Software, Validation, Visualization, Writing – original draft, Writing – review and editing | Charlyn Shue, Data curation, Investigation, Methodology, Writing – review and editing | W. Matthew Henderson, Data curation, Formal analysis, Investigation, Methodology, Project administration, Resources, Software, Supervision, Validation, Writing – review and editing | Donna A. Glinski, Data curation, Formal analysis, Investigation, Methodology, Project administration, Resources, Software, Supervision, Validation, Writing – review and editing | Erin K. Lipp, Conceptualization, Funding acquisition, Methodology, Project administration, Resources, Software, Supervision, Validation, Writing – review and editing

## DATA AVAILABILITY

Metabolomics data collected from iron and starvation comparisons can be found in the supplemental material under Tables S1 and S2.

## ADDITIONAL FILES

The following material is available online.

### Supplemental Material

**Data Set S1 (Spectrum02680-23-s0001.xlsx).** Growth kinetics summary data.
**Supplemental material (Spectrum02680-23-s0002.docx).** Fig. S1 to S11; Tables S3 and S4.
**Tables S1 and S2 (Spectrum02680-23-s0003.xlsx).** Significantly upregulated *V. alginolyticus* endometabolites detected during iron supplementation and iron starvation experiments.

### Open Peer Review

**PEER REVIEW HISTORY (review-history.pdf).** An accounting of the reviewer comments and feedback.

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
