## [Reviewer comments · Microbiology Spectrum]

Microbiology Spectrum

***Vibrio alginolyticus* growth kinetics and the metabolic effects of iron.**

William Norfolk, Charlyn Shue, William Henderson, Donna Glinski, and Erin Lipp

Corresponding Author(s): Erin Lipp, University of Georgia

Review Timeline:

Submission Date:	July 26, 2023
Editorial Decision:	September 8, 2023
Revision Received:	October 2, 2023
Accepted:	October 11, 2023

Editor: Tino Polen

Reviewer(s): Disclosure of reviewer identity is with reference to reviewer comments included in decision letter(s). The following individuals involved in review of your submission have agreed to reveal their identity: Yibei Zhang (Reviewer #1)

Transaction Report:

DOI: <https://doi.org/10.1128/spectrum.02680-23>

September 8, 2023

Dr. Erin K Lipp
University of Georgia
Dept. of Environmental Health Science
206 Environmental Health Science Bldg.
Athens, GA 30602-2102

Re: Spectrum02680-23 (*Vibrio alginolyticus* growth kinetics and the metabolic effects of iron.)

Dear Dr. William Anderson Norfolk,
Dear Dr. Erin K. Lipp,

thank you for submitting your manuscript to Microbiology Spectrum.

I received comments on your manuscript from two expert reviewers. Both reviewers suggested modifications to improve the manuscript.

I do hope you find the reviewers' comments below helpful and look forward to receiving a revised version from you.

Link Not Available

Sincerely,
Tino Polen
Editor, Microbiology Spectrum

Journals Department
Reviewer comments:

Reviewer #1 (Comments for the Author):

This manuscript evaluated the effects of temperature, salinity, and iron condition on the growth response in *V. alginolyticus*. Moreover, the metabolic effects of iron supplementation was investigated by GCMS. The authors found that *V. alginolyticus* strains are capable of rapid growth under a broad range of favorable temperature and salinity levels, which can be affected by the presence of iron. Overall, this is a comprehensive work, which is worthy for publication. Some comments are as follows:

- 1.Line 35 and 327, what does the "Fe" indicated?ferric iron or ferrous iron or the total of iron?
- 2.Line 133, "Scavenging of host-derived iron leads to increased bacterial replication resulting in *V. alginolyticus* persistence and

increased severity of infection (Kustus et al., 2011)". Is this view right? I did not find this opinion in the review article written by Kustus et al., (2011). In addition, the low iron levels have been shown to induce the expression of a number of bacterial toxins and virulence factors (PMID: 16912433). And many researchers verified that iron promotes biofilm formation in vivo and in vitro (PMID: 36383258; PMID: 35410114). So I think the author should reconsider the description here.

3. Figure 1, please use the integrated name of strains instead of abbreviation.

4. Line 218, SFE+ should be SFe+

5. The conclusion can be condensed.

Reviewer #2 (Comments for the Author):

Norfolk et al. perform a study on the *Vibrio alginolyticus* growth kinetics and the metabolic effects of iron. Authors used growth conditions, and metabolomics to analyze the effects of temperature, salinity, and iron on vibrio survival and growth. The study is interesting and merits publication, however, I have a number of remarks that require attention.

1-Please confirm that OD derived counts match cell cytometry or CFU counts, or cell microscopy counts. The numbers should be similar.

2-Please confirm by bioinformatics tools that the genome sequences of the studied strains reflect the temperature and salinity ranges obtained in the in vitro tests. The authors may use a bioinfo tool to verify congruence between their experiments and data retrieved directly from genome sequences, the so called in silico phenotypes. 10.1186/1471-2164-13-S7-S3 , 10.1093/bioinformatics/btz059

3-why did you choose 4µM of Iron for the metabolomics? Also it is unclear how these vibrios would tolerate higher amounts of iron in the media. This is relevant in the face of possible iron inputs in the ocean and on coral reefs. *Alginolyticus* may appear in the water column in some occasions. <https://peerj.com/articles/741/> , 10.1126/science.1106028 . High levels of Iron might induce the growth of putative vibrios in the water 10.1016/j.scitotenv.2019.135914 , 10.1016/j.scitotenv.2018.11.112 which in turn might kill corals . Is 4 µM much in terms of possible marine systems vibrio may play a role? Ex bays, reefs. These concentrations of iron are common in such systems?

4-the metabolomics part is preliminary? I had the impression the analyses could provide the specific molecules and metabolite names induced by iron. Not only the general kegg groups. Also match metabolome data with genome data to elucidate gene function and in silico phenotyping. I do not think the so called unique metabolic pathways were clear enough and they require molecule IDs. For instance, which terpenoids and polyketides were identified?

Minor remarks

-Impossible to see a blue line (ATCC) in fig3A.

-as a study control, the authors could have used *Vibrio parahaemolyticus* which is a sister species of *V. alginolyticus*. The comparison would be useful for many reasons.

-please improve conclusions and avoid results now.

Staff Comments:

Preparing Revision Guidelines

Please return the manuscript within 60 days; if you cannot complete the modification within this time period, please contact me. If you do not wish to modify the manuscript and prefer to submit it to another journal, please notify me of your decision immediately so that the manuscript may be formally withdrawn from consideration by Microbiology Spectrum.

Response to Reviewers
(author's responses are in red)

Reviewer #1 (Comments for the Author):

This manuscript evaluated the effects of temperature, salinity, and iron condition on the growth response in *V. alginolyticus*. Moreover, the metabolic effects of iron supplementation was investigated by GCMS. The authors found that *V. alginolyticus* strains are capable of rapid growth under a broad range of favorable temperature and salinity levels, which can be affected by the presence of iron. Overall, this is a comprehensive work, which is worthy for publication. Some comments are as follows:

1. Line 35 and 327, what does the "Fe" indicated? ferric iron or ferrous iron or the total of iron?
Thank you for your comment. These levels are indicative of ferric iron in the media added as designated FeCl₃ amendments. We have modified word use in the abstract as well as added additional information to the methods section to clarify these concentrations.

Lines: 481-486. "Prior work indicates that LB provides ~17 μM Fe (Abdul-Tehrani et al., 1999). Iron effects were measured between 0.2 and 20 μM Fe(III) at 0.2, 0.5, 1, 3, 4, 10, and 20 μM added FeCl₃. These levels were selected to represent a range of iron concentrations from environmentally relevant (0.2-4 μM) to highly elevated levels (10-20 μM). Iron concentrations above 20 μM were not evaluated due to the formation of precipitate which interfered with the accurate collection of optical density measures."

2. Line 133, "Scavenging of host-derived iron leads to increased bacterial replication resulting in *V. alginolyticus* persistence and increased severity of infection (Kustusch et al., 2011)". Is this view right? I did not find this opinion in the review article wrote by Kustusch et al., (2011). In addition, the low iron levels have been shown to induce the expression of a number of bacterial toxins and virulence factors (PMID: 16912433). And many researchers verified that iron promote biofilm formation in vivo and in vitro (PMID: 36383258; PMID: 35410114). So I think the author should reconsider the description here.

Thank you for your comment. We have amended the wording of this section of the introduction and added additional references citing the importance of iron for *Vibrio* spp. virulence.

Lines: 129-136. "While these systems enhance the competitiveness of *V. alginolyticus* in environmental settings, they also contribute to its establishment during infection by outcompeting host iron sequestration mechanisms or directly scavenging iron from heme in blood cells thus, increasing the iron pool available to infecting cells (Wang et al., 2008; Kustusch et al., 2011). Increased iron availability is known to increase bacterial replication (Wright et al., 1981) and promote biofilm formation (Çam & Brinkmeyer, 2019) in *Vibrio* spp. which can contribute to the onset and severity of infection."

3. Figure 1, please use the integrated name of strains instead of abbreviation.

Thank you for pointing out this inconsistency, we have replaced the legend names for figures 1-3 to match the full names in text.

4.Line 218, SFE+ should be SFe+
Thank you, we have corrected this typo.

5.The conclusion can be condensed.
Thank you for your comment. We have condensed the language of the conclusion to cover the broad findings of this work.

Reviewer #2 (Comments for the Author):

Norfolk et al. perform a study on the *Vibrio alginolyticus* growth kinetics and the metabolic effects of iron. Authors used growth conditions, and metabolomics to analyze the effects of temperature, salinity, and iron on vibrio survival and growth. The study is interesting and merits publication, however, I have a number of remarks that require attention.

1-Please confirm that OD derived counts match cell cytometry or CFU counts, or cell microscopy counts. The numbers should be similar.

Optical density measurements were employed in this study to evaluate the growth kinetics of *V. alginolyticus* strains under the designated temperature, salinity, and iron conditions. Culture-based CFU counts were only used to quantify cellular growth for the metabolomics portion of the experiment prior to processing with gas chromatography mass spectrometry; thus, these values were not compared as they originate from different samples.

2-Please confirm by bioinformatics tools that the genome sequences of the studied strains reflect the temperature and salinity ranges obtained in the in vitro tests. The authors may use a bioinfo tool to verify congruence between their experiments and data retrieved directly from genome sequences, the so called in silico phenotypes. 10.1186/1471-2164-13-S7-S3 , 10.1093/bioinformatics/btz059

Thank you for your comment. At current, only strain ATCC 17749 has a fully available genome for comparison using these methods. Furthermore, the intention of this research was to demonstrate the importance of temperature, salinity, and iron content together as determinants for *V. alginolyticus* growth. While the methods of *in silico* phenotyping are an interesting avenue to pursue – to the best of our knowledge, these methods have not been validated for salinity and iron tolerance in prokaryotes and thus are beyond the scope of this research.

3-why did you chose 4uM of Iron for the metabolomics?

The use of 4 μ M iron for metabolomics experiments was selected to create an iron concentration consistent with levels observed on the upper end of normal for coastal systems. We have added additional context to the methods section to clarify this.

Lines: 538-541. “4 μ M was designated as iron replete based on growth experiments for JW16-551 and prior work by Westrich et al. (2016). This level is also consistent with iron concentrations found in non-oligotrophic coastal waters (Zhu et al., 2018), offering an environmentally relevant concentration that would be sufficient for growth.”

Also it is unclear if how these vibrios would tolerate higher amounts of iron in the media. This is relevant in the face of possible iron inputs in the ocean and on coral reefs. *Alginolyticus* may appear in the water column in some occasions. <https://peerj.com/articles/741/>, 10.1126/science.1106028 .

As stated in the results at lines 206-207, no growth limitation was observed for strains JW16-551 and JW16-580 at higher levels of iron suggesting that *V. alginolyticus* is able to tolerate iron concentrations up to 20 μ M iron.

High levels of Iron might induce the growth of putative vibrios in the water 10.1016/j.scitotenv.2019.135914 , 10.1016/j.scitotenv.2018.11.112 which in turn might kill corals.

We have added 10.1016/j.scitotenv.2019.135914 to our citations in the discussion at Lines 304-305.

Is 4 μ M much in terms of possible marine systems vibrio may play a role? Ex bays, reefs. This concentrations of iron are common in such systems?

Thank you for your comment. As stated above, an iron concentration of 4 μ M is at the upper end of normal for most coastal systems. However, the type of system can substantially impact the typical iron concentration. Offshore oligotrophic systems such as coral reefs are typically highly iron deficient (nanomolar level) whereas estuarine systems are less limited (micromolar level) due to riverine input. We have added additional context to the discussion and methods section to clarify our iron selections and provide environmental and methodological context.

Lines 340-343. “This finding is particularly important for oligotrophic systems such as coral reefs which typically possess lower iron pools (Entsch et al., 1983; Kelly et al., 2011) compared to estuarine systems (Boyle & Edmond, 1977; Sunda, 2012) and may be more prone to the formation of iron-induced *Vibrio* spp. blooms.”

Lines 481-486. Noted above under reviewer one comment #1.

4-the metabolomics part is preliminar? I had the impression the analyzes could provide the specific molecules and metabolites names induced by iron. Not only the general keeg groups. Also match metabolome data with genome data to elucidate gene function and in silico phenotyping. I do not think the so called unique metabolic pathways was clear enough and they require molecule ids. For instance, which terpenoids and polyketides were identified?

Thank you for your comment. The specific molecules identified from the metabolomics portion of this research were identified and reported (both names and HMDB numbers) for each experimental condition in tables S1 and S2. Additionally, the use of KEGG metabolic pathways as the major result endpoint was selected based on previously published studies using untargeted metabolomics to assess metabolic function (<https://doi.org/10.1128/spectrum.02067-22>; <https://doi.org/10.1128/Spectrum.00625-21>; <https://doi.org/10.1128/aac.02109-15>). Furthermore, many metabolites identified in this study have the potential to function in multiple metabolic pathways thus, reporting based on pathway enrichment considers all possible pathways that may be represented by the specific chemical species identified.

Minor remarks

-Impossible to see a blue line (ATCC) in fig3A.

The blue line representing ATCC strain 17749 is shown in figure 3A under the 20 μ M iron treatment only. This is due to the fact that the iron limiting media utilized in this experiment severely limited the growth of this strain at all iron concentrations – with 20 μ M as the only concentration with sufficient growth to quantify doubling time. As such, the blue line only appears as a single linerange value under this treatment representing the standard error of growth for this strain. An explanation of this limitation can be found in the in the figure captions at lines 872-876.

-as a study control, the authors could have used *Vibrio parahaemolyticus* which is a sister species of *V. alginolyticus*. The comparison would be useful for many reasons.

We appreciate that tests on additional *Vibrio* species would be informative, especially the closely related *V. parahaemolyticus*, but the intention of our study was to evaluate the abiotic tolerance parameters and importance of iron to *V. alginolyticus*, as this species is highly understudied compared to other non-Cholera vibrios.

-please improve conclusions and avoid results now.

Addressed above in comment #5 from reviewer one.

October 11, 2023

Dr. Erin K Lipp
University of Georgia
Dept. of Environmental Health Science
206 Environmental Health Science Bldg.
Athens, GA 30602-2102

Re: Spectrum02680-23R1 (*Vibrio alginolyticus* growth kinetics and the metabolic effects of iron.)

Dear Dr. William Anderson Norfolk,
Dear Dr. Erin K. Lipp,

your revised manuscript has been accepted, and I am forwarding it to the ASM Journals Department for publication. You will be notified when your proofs are ready to be viewed.

Sincerely,
Tino Polen
Editor, Microbiology Spectrum
